# Robust identification of perturbed cell types in single-cell RNA-seq data

Phillip B. Nicol[1], Danielle Paulson[1], Gege Qian[2], X. Shirley Liu [3],
Rafael Irizarry [3] ✉ & Avinash D. Sahu [4] ✉

Single-cell transcriptomics has emerged as a powerful tool for understanding how different cells contribute to disease progression by identifying cell types that change across diseases or conditions. However, detecting changing cell types is challenging due to individual-to-individual and cohort-to-cohort variability and naive approaches based on current computational tools lead to false positive findings. To address this, we propose a computational tool, *scDist*, based on a mixed-effects model that provides a statistically rigorous and computationally efficient approach for detecting transcriptomic differences. By accurately recapitulating known immune cell relationships and mitigating false positives induced by individual and cohort variation, we demonstrate that *scDist* outperforms current methods in both simulated and real datasets, even with limited sample sizes. Through the analysis of COVID-19 and immunotherapy datasets, *scDist* uncovers transcriptomic perturbations in dendritic cells, plasmacytoid dendritic cells, and FCER1G+NK cells, that provide new insights into disease mechanisms and treatment responses. As single-cell datasets continue to expand, our faster and statistically rigorous method offers a robust and versatile tool for a wide range of research and clinical applications, enabling the investigation of cellular perturbations with implications for human health and disease.

The advent of single-cell technologies has enabled measuring transcriptomic profiles at single-cell resolution, paving the way for the identification of subsets of cells with transcriptomic profiles that differ across conditions. These cutting-edge technologies empower researchers and clinicians to study human cell types impacted by drug treatments, infections like SARS-CoV-2, or diseases like cancer. To conduct such studies, scientists must compare single-cell RNA-seq (scRNA-seq) data between two or more groups or conditions, such as infected versus non-infected[1], responders versus non-responders to treatment[2], or treatment versus control in controlled experiments.

Two related but distinct classes of approaches exist for comparing conditions in single-cell data: differential abundance prediction and differential state analysis[3]. Differential abundance approaches,

such as DA-seq, Milo, and Meld[4–7], focus on identifying cell types with varying proportions between conditions. In contrast, differential state analysis seeks to detect predefined cell types with distinct transcriptomic profiles between conditions. In this study, we focus on the problem of differential state analysis.

Past differential state studies have relied on manual approaches involving visually inspecting data summaries to detect differences in scRNA data. Specifically, cells were clustered based on gene expression data and visualized using uniform manifold approximation (UMAP)[8]. Cell types that appeared separated between the two conditions were identified as different[1]. Another common approach is to use the number of differentially expressed genes (DEGs) as a metric for transcriptomic perturbation. However, as noted by ref. 9, the number of

[1]Harvard University, Cambridge, MA, USA. [2]University of California San Diego School of Medicine, San Diego, CA, USA. [3]Dana-Farber Cancer Institute, Boston, MA, USA. [4]University of New Mexico Comprehensive Cancer Center, Albuquerque, NM, USA. ✉e-mail: rafael_irizarry@dfci.harvard.edu; asahu@salud.unm.edu

DEGs depends on the chosen significance level and can be confounded by the number of cells per cell type because this influences the power of the corresponding statistical test. Additionally, this approach does not distinguish between genes with large and small (yet significant) effect sizes.

To overcome these limitations, *Augur*[9] uses a machine learning approach to quantify the cell-type specific separation between the two conditions. Specifically, *Augur* trains a classifier to predict condition labels from the expression data and then uses the area under the receiver operating characteristic (AUC) as a metric to rank cell types by their condition difference. However, *Augur* does not account for individual-to-individual variability (or pseudoreplication[10]), which we show can confound the rankings of perturbed cell types.

In this study, we develop a statistical approach that quantifies transcriptomic shifts by estimating the distance (in gene expression space) between the condition means. This method, which we call *scDist*, introduces an interpretable metric for comparing different cell types while accounting for individual-to-individual and technical variability in scRNA-seq data using linear mixed-effect models. Furthermore, because transcriptomic profiles are high-dimensional, we develop an approximation for the between-group differences, based on a low-dimensional embedding, which results in a computationally convenient implementation that is substantially faster than *Augur*. We demonstrate the benefits using a COVID-19 dataset, showing that *scDist* can recover biologically relevant between-group differences while also controlling for sample-level variability. Furthermore, we demonstrated the utility of the *scDist* by jointly inferring information from five single-cell immunotherapy cohorts, revealing significant differences in a subpopulation of NK cells between immunotherapy responders and non-responders, which we validated in bulk transcriptomes from 789 patients. These results highlight the importance of accounting for individual-to-individual and technical variability for robust inference from single-cell data.

## Results

### Not accounting for individual-to-individual variability leads to false positives

We used blood scRNA-seq from six healthy controls[1] (see Table 1), and randomly divided them into two groups of three, generating a negative control dataset in which no cell type should be detected as being different. We then applied *Augur* to these data. This procedure was repeated 20 times. *Augur* falsely identified several cell types as perturbed (Fig. 1A). *Augur* quantifies differences between conditions with an AUC summary statistic, related to the amount of transcriptional separation between the two groups (AUC = 0.5 represents no difference). Across the 20 negative control repeats, 93% of the AUCs (across all cell typess) were >0.5, and red blood cells (RBCs) were identified as perturbed in all 20 trials (Fig. 1A). This false positive result was in part

due to high across-individual variability in cell types such as RBCs (Fig. 1B).

We confirmed that individual-to-individual variation underlies false positive predictions made by *Augur* using a simulation. We generated simulated scRNA-seq data with no condition-level difference and varying patient-level variability (Methods). As patient-level variability increased, differences estimated by *Augur* also increased, converging to the maximum possible AUC of 1 (Fig. 1C): *Augur* falsely interpreted individual-to-individual variability as differences between conditions.

*Augur* recommends that unwanted variability should be removed in a pre-processing step using batch correction software. We applied Harmony[11] to the same dataset[1], treating each patient as a batch. We then applied *Augur* to the resulting batch corrected PC scores and found that several cell types still had AUCs significantly above the null value of 0.5 (Fig. S1a). On simulated data, batch correction as a pre-processing step also leads to confounding individual-to-individual variability as condition difference (Fig. S1b).

### A model-based distance metric controls for false positives

To account for individual-to-individual variability, we modeled the vector of normalized counts with a linear mixed-effects model. Mixed models have previously been shown to be successful at adjusting for this source of variability[10]. Specifically, for a given cell type, let $\mathbf{z}_{ij}$ be a length $G$ vector of normalized counts for cell $i$ and sample $j$ ($G$ is the number of genes). We then model

$$\mathbf{z}_{ij} = \boldsymbol{\alpha} + x_j\boldsymbol{\beta} + \boldsymbol{\omega}_j + \boldsymbol{\varepsilon}_{ij} \tag{1}$$

where $\boldsymbol{\alpha}$ is a vector with entries $\alpha_g$ representing the baseline expression for gene $g$, $x_j$ is a binary indicator that is 0 if individual $j$ is in the reference condition, and 1 if in the alternative condition, $\boldsymbol{\beta}$ is a vector with entries $\beta_g$ representing the difference between condition means for gene $g$, $\boldsymbol{\omega}_j$ is a random effect that represents the differences between individuals, and $\boldsymbol{\varepsilon}_{ij}$ is a random vector (of length $G$) that accounts for other sources of variability. We assume that $\boldsymbol{\omega}_j \overset{\text{i.i.d}}{\sim} \mathcal{N}(0,\tau^2 I)$, $\boldsymbol{\varepsilon}_{ij} \overset{\text{i.i.d}}{\sim} \mathcal{N}(0,\sigma^2 I)$, and that the $\boldsymbol{\omega}_j$ and $\boldsymbol{\varepsilon}_{ij}$ are independent of each other.

To obtain normalized counts, we recommend defining $z_{ij}$ to be the vector of Pearson residuals obtained from fitting a Poisson or negative binomial GLM[12], the normalization procedure is implemented in the scTransform function[13]. However, our proposed approach can be used with other normalization methods for which the model is appropriate.

Note that in model (18), the means for the two conditions are $\alpha$ and $\boldsymbol{\alpha} + \boldsymbol{\beta}$, respectively. Therefore, we quantify the difference in expression profile by taking the 2 − norm of the vector $\beta$:

$$D := \left\|\boldsymbol{\beta}\right\|_2 = \left(\boldsymbol{\beta}^\top \boldsymbol{\beta}\right)^{1/2} = \sqrt{\sum_{g=1}^{G}\beta_g^2}. \tag{2}$$

Here, $D$ can be interpreted as the Euclidean distance between condition means (Fig. 2A).

Because we expected the vector of condition differences $\beta$ to be sparse, we improved computational efficiency by approximating $D$ with a singular value decomposition to find a $K \times G$ matrix $U$, with $K$ much smaller than $G$, and

$$D \approx D_K := \sqrt{\sum_{k=1}^{K}(U\boldsymbol{\beta})_k^2}. $$

With this approximation in place, we fitted model equation (18) by replacing $z_{ij}$ with $Uz_{ij}$ to obtain estimates of $(U\beta)_k$. A challenge with estimating $D_K$ is that the maximum likelihood estimator can have a significant upward bias when the number of patients is small (as is typically the case). For this reason, we employed a post-

**Table 1 | Datasets used in the figures**

| Figure | Dataset |
|---|---|
| Fig. 1 | Small COVID-19[1], Simulated null data |
| Fig. 3 | Small COVID-19[1], Simulated null data |
| Fig. 4 | Small COVID-19[1], Sorted immune cells[15,16] |
| Fig. 5 | Large COVID-19[17] |
| Fig. 6 | Immunotherapy cohorts:<br>• 4 scRNA datasets (discovery)[2,23-25]<br>• bulk RNA-Seq datasets (validation)[32-38] |

The small COVID-19 dataset was generated by ref. 1 and can be downloaded as a Seurat object from www.covid19cellatlas.org. The simulated null data can be generated using the *scDist* R package. The large COVID-19 was generated by ref. 17 and can be accessed from the respective publication. Immunotherapy cohorts used in Fig. 6 perform integrative analysis of 4 scRNA datasets for discovery, and for validation, 7 bulk RNA-Seq datasets are utilized.

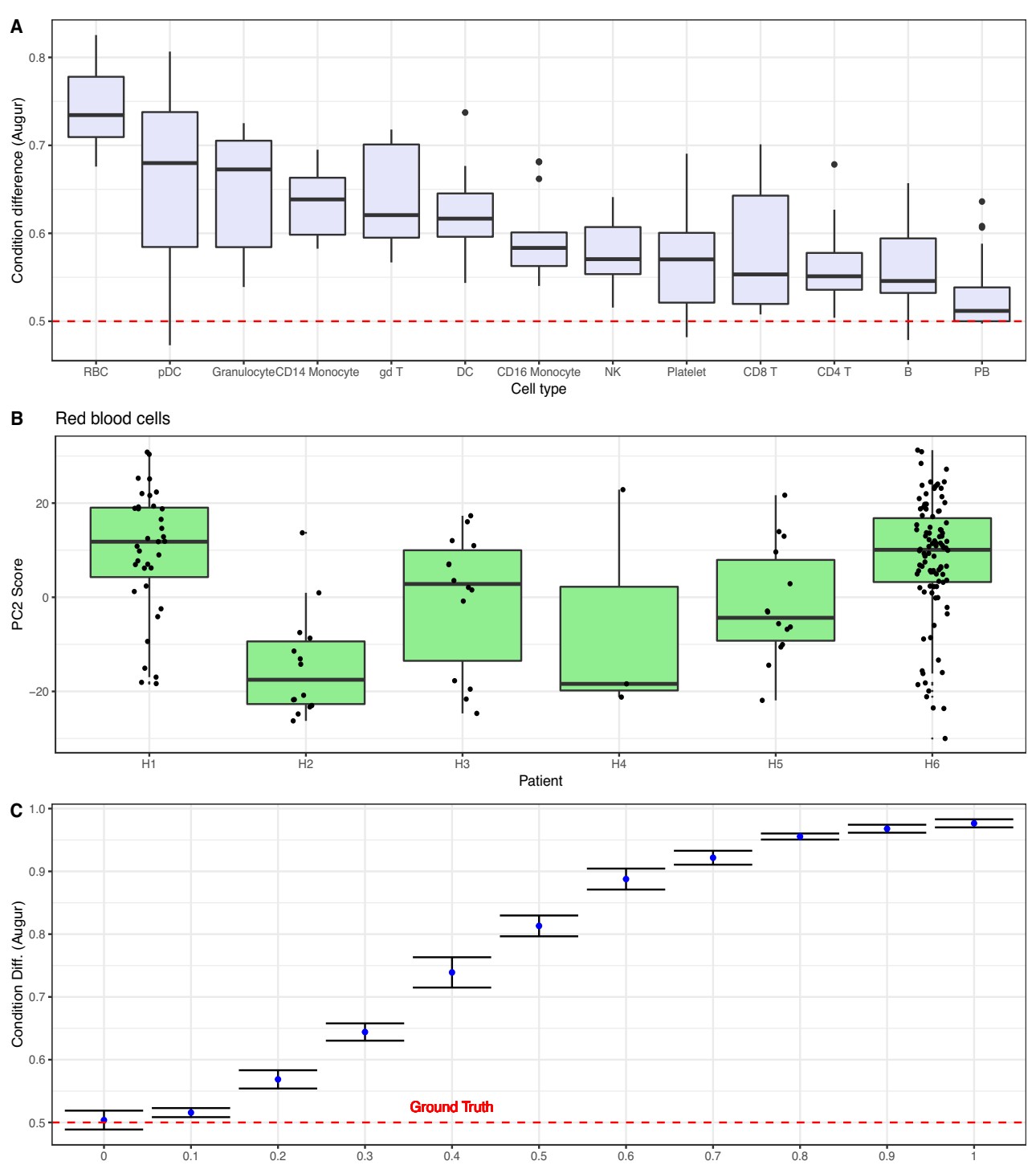

**Fig. 1 | Evaluating *Augur*'s performance in negative control experiments.**
**A** AUCs achieved by *Augur* on 20 random partitions of healthy controls (*n* = 6 total patients divided randomly into two groups of 3), with no expected cell type differences (dashed red line indicates the null value of 0.5). **B** Boxplot depicting the second PC score for red blood cells from healthy individuals, highlighting high across-individual variability (each box represents a different individual). The boxplots display the median and first/third quartiles. **C** AUCs achieved by *Augur* on simulated scRNA-seq data (10 individuals, 50 cells per individual) with no condition differences but varying patient-level variability (dashed red line indicates the ground truth value of no condition difference, AUC 0.5), illustrating the influence of individual-to-individual variability on false positive predictions. Points and error bands represent the mean ±1 SD. Source data are provided as a Source Data file.

hoc Bayesian procedure to shrink $(U\boldsymbol{\beta})_k^2$ towards zero and compute a posterior distribution of $D_K$[14]. We also provided a statistical test for the null hypothesis that $D_K = 0$. We refer to the resulting procedure as *scDist* (Fig. 2B). Technical details are provided in Methods.

We applied *scDist* to the negative control dataset based on blood scRNA-seq from six healthy used to show the large number of false positives reported by *Augur* (Fig. 1) and found that the false positive rate was controlled (Fig. 3A, B). We then applied *scDist* to the data from the simulation study and found that, unlike *Augur*, the resulting

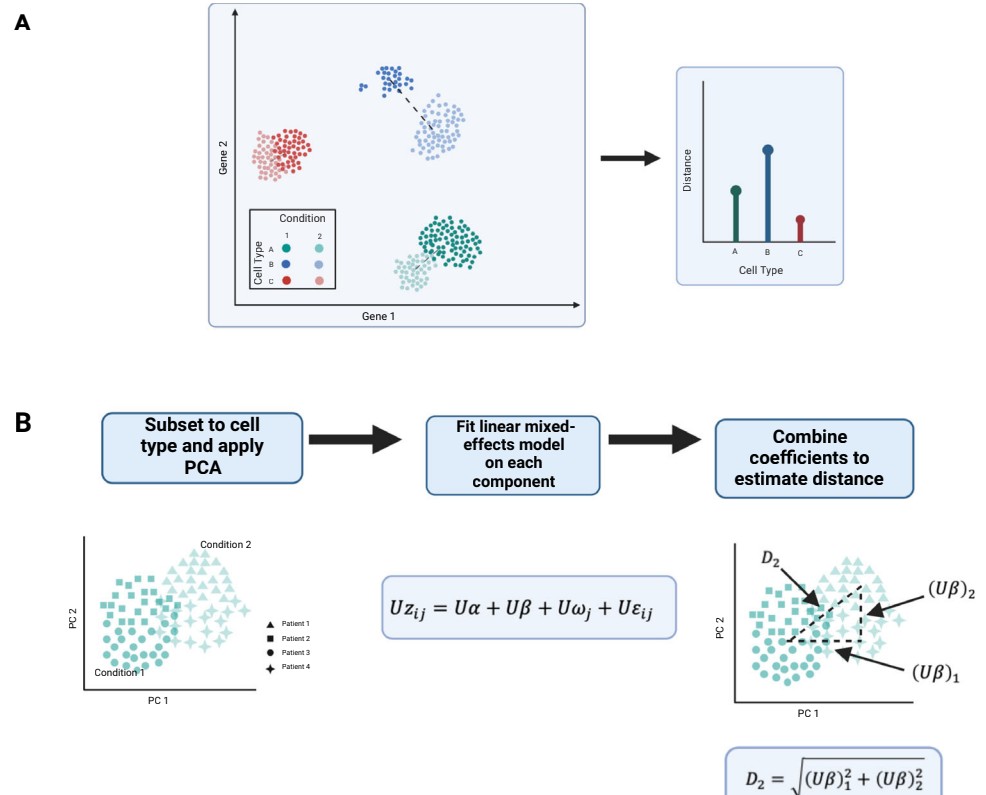

**Fig. 2 | Visual representation of the *scDist* method. A** *scDist* estimates the distance between condition means in high-dimensional gene expression space for each cell type. **B** To improve efficiency, *scDist* calculates the distance in a low-dimensional embedding space (derived from PCA) and employs a linear mixed-effects model to account for sample-level and other technical variability. This figure is created with Biorender.com, was released under a Creative Commons Attribution-NonCommercial-NoDerivs 4.0 International license.

distance estimate does not grow with individual-to-individual variability (Fig. 3C). *scDist* also accurately estimated distances on fully simulated data (Fig. S2).

The Euclidean distance $D$ measures perturbation by taking the sum of squared differences across all genes. To show that this measure is biologically meaningful, we applied *scDist* to obtain estimated distances between pairs of known cell types in the above dataset and then applied hierarchical clustering to these distances. The resulting clustering is consistent with known relationships driven by cell lineages (Fig. 3D). Specifically, Lymphoid cell types T and NK cells clustered together, while B cells were further apart, and Myeloid cell types DC, monocytes, and neutrophils were close to each other.

Though the *scDist* distance $D$ assigns each gene an equal weight (unweighted), scDist includes an option to assign different weights $w_g$ to each gene (Methods). Weighting could be useful in situations where certain genes are known to contribute more to specific phenotypes. We conducted a simulation to study the impact of using the weighted distance. These simulations show that when a priori information is available, using the correct weighting leads to a slightly better estimation of the distance. However, incorrect weighting leads to significantly worse estimation compared to the unweighted distance (Fig. S3). Therefore, the unweighted distance is recommended unless strong a priori information is available.

Challenges in cell type annotations are expected to impact *scDist*'s interpretation, much like it does for other methods reliant on a priori cell type annotation such as[3,9]. Our simulations (see Methods), reveal *scDist*'s vulnerability to false-negatives when annotations are confounded by condition- or patient-specific factors. However, when clusters are annotated using data where such differences have been removed, scDist's predictions become more reliable (Fig. S23). Thus,

we recommend removing these confounders before annotation. As potential issues could occur when the inter-condition distance exceeds the inter-cell-type distance, *scDist* provides a diagnostic plot (Fig. S6) to compare these two distances. *scDist* also incorporates an additional diagnostic feature (Fig. S24) to identify annotation issues, utilizing a cell-type tree to evaluate cell relationships at different hierarchical levels. Inconsistencies in *scDist*'s output signal potential clustering or annotation errors.

**Comparison to counting the number of DEGs**
We also compared *scDist* to the approach of counting the number of differentially expressed genes (nDEG) on pseudobulk samples[3]. Given that the statistical power to detect DEGs is heavily reliant on sample size, we hypothesized that nDEG could become a misleading measure of perturbation in single-cell data with a large variance in the number of cells per cell type. To demonstrate this, we applied both methods to resampled COVID-19 data[1] where the number of cells per cell type was artificially varied between 100 and 10,000. nDEG was highly confounded by the number of cells (Fig. 4A), whereas the *scDist* distance remained relatively constant despite the varying number of cells (Fig. 4B). When the number of subsampled cells is small, the ranking of cell types (by perturbation) was preserved by *scDist* but not by nDEG (Fig. S5a–c). Additionally, *scDist* was over 60 times faster than nDEG since the latter requires testing all $G$ genes as opposed to $K \ll G$ PCs (Fig. S4).

An additional limitation of nDEG is that it does not account for the magnitude of the differential expression. We illustrated this with a simple simulation that shows the number of DEGs between two cell types can be the same (or less) despite a larger transcriptomic perturbation in gene expression space (Fig. S7a, b). To demonstrate this

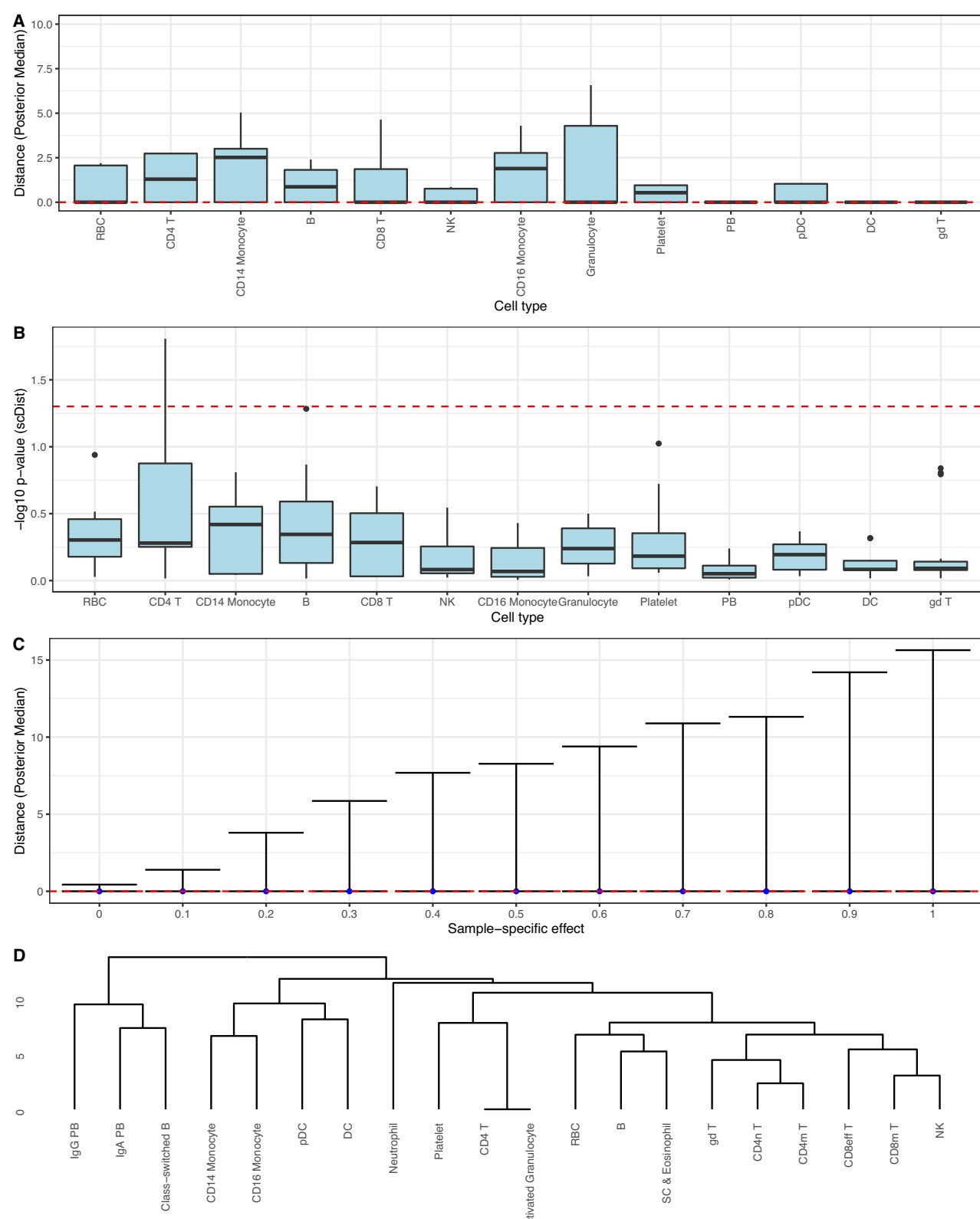

**Fig. 3 | Application and performance of *scDist*. A** Reanalysis of the data from Fig. 1A using distances calculated with *scDist*; the dashed red line represents ground truth value of 0. **B** As in **A**, but for p-values computed with *scDist*; values above the dashed red line represent *p* < 0.05. **C** Null simulation from Fig. 1C reanalyzed using distances calculated by *scDist*; the dashed red line represents the ground truth value of 0. Points and error bands represent the mean ±1 SD. **D** Dendrogram generated by hierarchical clustering based on distances between pairs of cell types estimated by *scDist*. In all panels the boxplots display median and first/third quartiles. Source data are provided as a Source Data file.

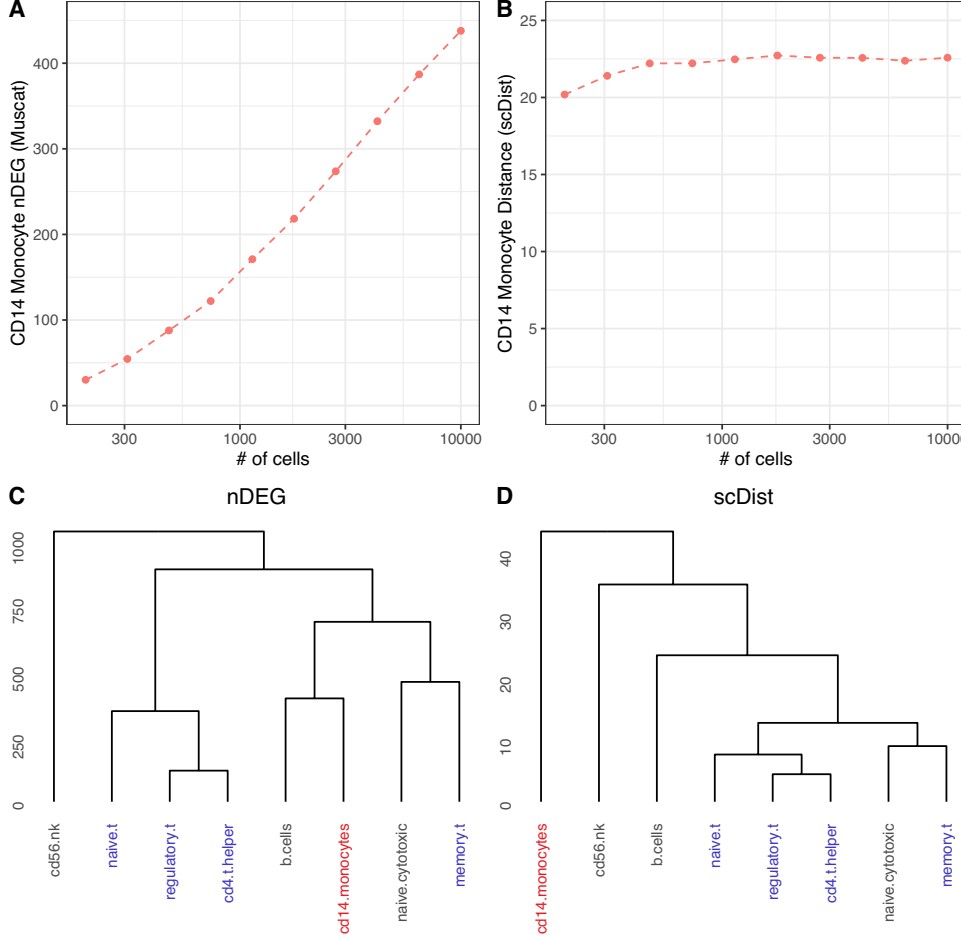

**Fig. 4 | The number of differentially expressed genes is susceptible to differences in statistical power. A** Sampling with replacement from the COVID-19 dataset[1] to create datasets with a fixed number of cells per cell type, and then counting the number of differentially expressed genes (nDEG) for the CD14 monocytes. **B** Repeating the previous analysis with the *scDist* distance.

**C** Comparing all pairs of cell types on the downsampled[15] dataset and applying hierarchical clustering to the pairwise perturbations. Leaves corresponding to T cells are colored blue while the leaf corresponding to monocytes is colored red. **D** The same analysis using the *scDist* distances. Source data are provided as a Source Data file.

on real data, we considered a dataset consisting of eight sorted immune cell types (originally from ref. 15 and combined by ref. 16) where *scDist* and nDEG were applied to all pairs of cell types, and the perturbation estimates were visualized using hierarchical clustering. Although both nDEG and *scDist* performed well when the sample size was balanced across cell types (Fig. S8), nDEG provided inconsistent results when the CD14 Monocytes were downsampled to create a heterogeneous cell type size distribution. Specifically, *scDist* produced the expected result of clustering the T cells together, whereas nDEG places the Monocytes in the same cluster as B and T cells (Fig. 4C, D) despite the fact that these belong to different lineages. Thus by taking into account the magnitude of the differential expression, *scDist* is able to produce results more in line with known biology.

We also considered varying the number of patients on simulated data with a known ground truth. Again, the nDEG (computed using a mixed model, as recommended by ref. 10) increases as the number of patients increases, whereas *scDist* remains relatively stable (Fig. S9a). Moreover, the correlation between the ground truth perturbation and *scDist* increases as the number of patients increases (Fig. S9b). Augur was also sensitive to the number of samples and had a lower correlation with the ground truth than both nDEG and *scDist*.

## *scDist* detects cell types that are different in COVID-19 patient compared to controls

We applied *scDist* to a large COVID-19 dataset[17] consisting of 1.4 million cells of 64 types from 284 PBMC samples from 196 individuals consisting of 171 COVID-19 patients and 25 healthy donors. The large number of samples of this dataset permitted further evaluation of our approach using real data rather than simulations. Specifically, we defined true distances between the two groups by computing the sum of squared log fold changes (across all genes) on the entire dataset and then estimated the distance on random samples of five cases versus five controls. Because *Augur* does not estimate distances explicitly, we assessed the two methods' ability to accurately recapitulate the ranking of cell types based on established ground truth distances. We found that *scDist* recovers the rankings better than *Augur* (Fig. 5A, S10). When the size of the subsample is increased to 15 patients per condition, the accuracy of *scDist* to recover the ground truth rank and distance improves further (Fig. S25).

To evaluate *scDist*'s accuracy further, we defined a new ground truth using the entire COVID-19 dataset, consisting two groups: four cell types with differences between groups (true positives) and five cell types without differences (false positives) (Fig. S11, Methods). We generated 1000 random samples with only five individuals per cell type and estimated group differences using both *Augur* and *scDist*.

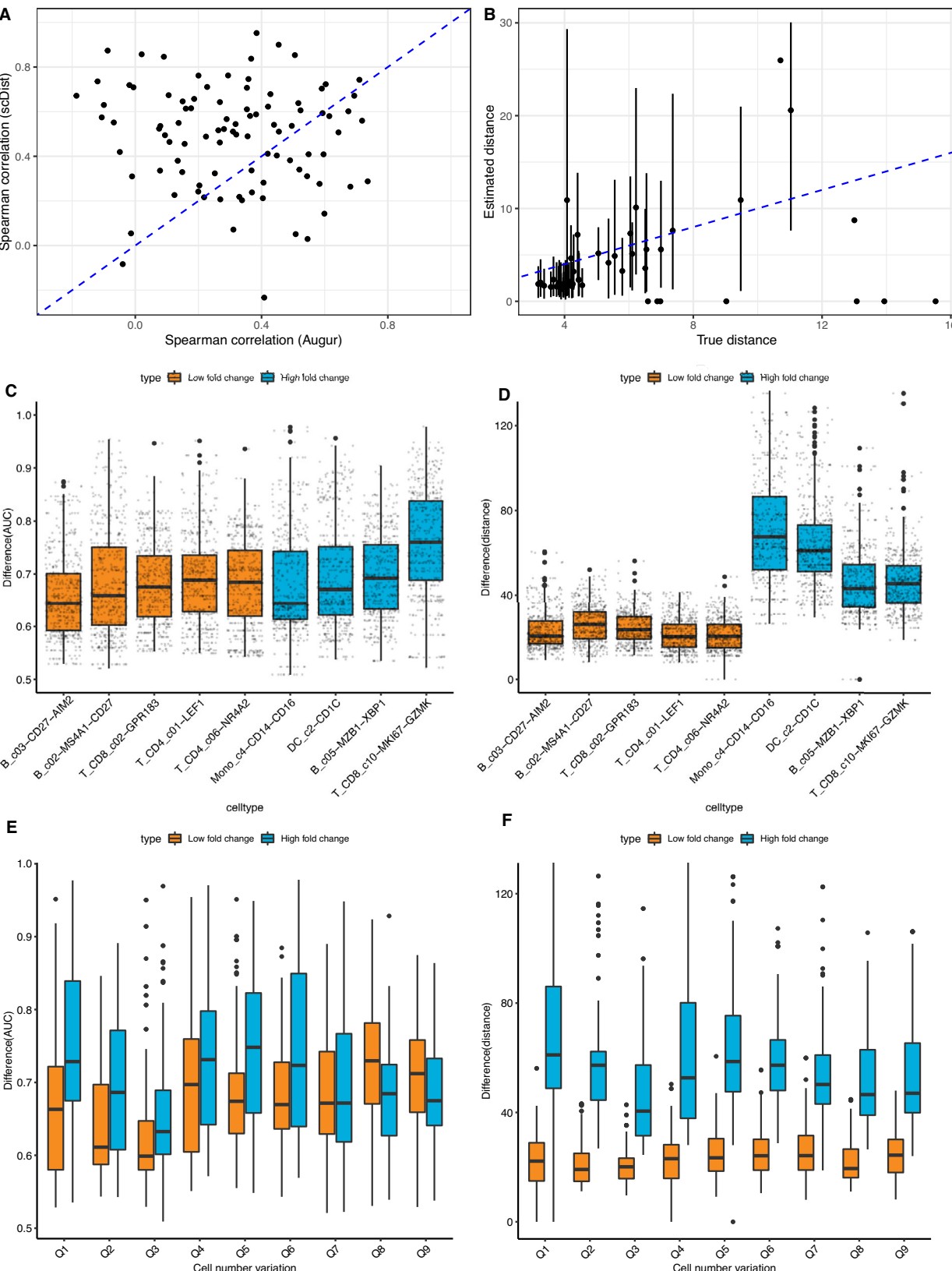

**Fig. 5 | Comparison of *scDist* and *Augur* performance based on real data simulation.** **A** Correlation between estimated ranks (based on subsamples of 5 cases and 5 controls) and true ranks for each method, with points above the diagonal line indicate better agreement of *scDist* with the true ranking. **B** Plot of true distance vs. distances estimated with *scDist* (dashed line represents *y* = *x*). Poins and error bars represent mean, and 5/95th percentile. **C** AUC values achieved by *Augur*, where color represents likely true (blue) or false (orange) positive cell types. **D** Same as **C**, but for distances estimated with *scDist*. **E** AUC values achieved by *Augur* against the cell number variation in subsampled-datasets (of false positive cell types). **F** Same as **E**, but for distances estimated with *scDist*. In all boxplots the median and first/third quartiles are reported. For **C**–**E**, 1000 random subsamples were used. Source data are provided as a Source Data file.

*Augur* failed to accurately separate the two groups (Fig. 5C); median difference estimates of all true positive cell types, except MK167+ CD8+T, were lower than median estimates of all true negative cell types (Fig. 5C). In contrast, *scDist* showed a separation between *scDist* estimates between the two groups (Fig. 5D).

Single-cell data can also exhibit dramatic sample-specific variation in the number of cells of specific cell types. This imbalance can arise from differences in collection strategies, biospecimen quality, and technical effects, and can impact the reliability of methods that do not account for sample-to-sample or individual-to-individual variation. We measured the variation in cell numbers within samples by calculating the ratio of the largest sample's cell count to the total cell counts across all samples (Methods). *Augur*'s predictions were negatively impacted by this cell number variation (Figs. 5E, S12), indicating its increased susceptibility to false positives when sample-specific cell number variation was present (Fig. 1C). In contrast, *scDist*'s estimates were robust to sample-specific cell number variation in single-cell data (Fig. 5F).

To further demonstrate the advantage of statistical inference in the presence of individual-to-individual variation, we analyzed the smaller COVID-19 dataset[1] with only 13 samples. The original study[1] discovered differences between cases and controls in CD14+ monocytes through extensive manual inspection. *scDist* identified this same group as the most significantly perturbed cell type. *scDist* also identified two cell types not considered in the original study, dendritic cells (DCs) and plasmacytoid dendritic cells (pDCs) ($p = 0.01$ and $p = 0.04$, Fig. S13a), although pDC did not remain significant after adjusting for multiple testing. We note that DCs induce anti-viral innate and adaptive responses through antigen presentation[18]. Our finding was consistent with studies reporting that DCs and pDCs are perturbed by COVID-19 infection[19,20]. In contrast, *Augur* identified RBCs, not CD14+ monocytes, as the most perturbed cell type (Fig. S14). Omitting the patient with the most RBCs dropped the perturbation between infected and control cases estimated by *Augur* for RBCs markedly (Fig. S14), further suggesting that *Augur* predictions are clouded by patient-level variability.

### *scDist* enables the identification of genes underlying cell-specific across-condition differences

To identify transcriptomic alteration, *scDist* assigns an importance score to each gene based on its contribution to the overall perturbation (Methods). We assessed this importance score for CD14+ monocytes in small COVID-19 datasets. In this cell type, *scDist* assigned the highest importance score to genes S100 calcium-binding protein A8 (*S100A8*) and S100 calcium-binding protein A9 (*S100A9*) ($p < 10^{-3}$, Fig. S13b). These genes are canonical markers of inflammation[21] that are upregulated during cytokine storm. Since patients with severe COVID-19 infections often experience cytokine storms, the result suggests that S100A8/A9 upregulation in CD14+ monocyte could be a marker of the cytokine storm[22]. These two genes were reported to be upregulated in COVID-19 patients in the study of 284 samples[17].

### *scDist* identifies transcriptomic alterations associated with immunotherapy response

To demonstrate the real-world impact of *scDist*, we applied it to four published dataset used to understand patient responses to cancer immunotherapy in head and neck, bladder, and skin cancer patients, respectively[2,23–25]. We found that each individual dataset was underpowered to detect differences between responders and non-responders (Fig. S15). To potentially increase power, we combined the data from all cohorts (Fig. 6A). However, we found that analyzing the combined data without accounting for cohort-specific variations led to false positives. For example, responder-non-responder differences estimated by *Augur* were highly correlated between pre- and post-treatments (Fig. 6B), suggesting a confounding effect of cohort-

specific variations. Furthermore, *Augur* predicted that most cell types were altered in both pre-treatment and post-treatment samples (AUC > 0.5 for 41 in pre-treatment and 44 in post-treatment out of a total of 49 cell types), which is potentially due to the confounding effect of cohort-specific variations.

To account for cohort-specific variations, we ran *scDist* including an explanatory variable to the model (18) to account for cohort effects. With this approach, distance estimates were not correlated significantly between pre- and post-treatment (Fig. 6B). Removal of these variables re-established correlation (Fig. S16). *scDist* predicted CD4-T and CD8-T altered pre-treatment (Fig. S17a), while NK, CD8-T, and B cells altered post-treatment (Fig. S17b). Analysis of subtypes revealed FCER1G+NK cells (NK-2) were changed in both pre-treatment and post-treatment samples (Fig. 6C). To validate this finding, we generated an NK-2 signature differential between responders and non-responders (Fig. S18 Methods) and evaluated these signatures in bulk RNA-seq immunotherapy cohorts, composing 789 patient samples (Fig. 6A). We scored each of the 789 patient samples using the NK-2 differential signature (Methods). The NK-2 signature scores were significantly associated with overall and progression-free survival (Fig. 6D) as well as radiology-based response (Fig. 6E). We similarly evaluated the top *Augur* prediction. Differential signature from plasma, the top predicted cell type by *Augur*, did not show an association with the response or survival outcomes in 789 bulk transcriptomes (Fig. S19, Methods).

### *scDist* is computationally efficient

A key strength of the linear modeling framework used by *scDist* is that it is efficient on large datasets. For instance, on the COVID-19 dataset with 13 samples[1], *scDist* completed the analysis in around 50 seconds, while *Augur* required 5 minutes. To better understand how runtime depends on the number of cells, we applied both methods to subsamples of the dataset that varied in size and observed that *scDist* was, on average, five-fold faster (Fig. S20). *scDist* is also capable of scaling to millions of cells. On simulated data, *scDist* required approximately 10 minutes to fit a dataset with 1,000,000 cells (Fig. S21). We also tested the sensitivity of *scDist* to the number of PCs used by comparing $D_K$ for various values of $K$. We observed that the estimated distances stabilize as $K$ increases (Fig. S22), justifying $K = 20$ as a reasonable choice for most datasets.

## Discussion

The identification of cell types influenced by infections, treatments, or biological conditions is crucial for understanding their impact on human health and disease. We present *scDist*, a statistically rigorous and computationally fast method for detecting cell-type specific differences across multiple groups or conditions. By using a mixed-effects model, *scDist* estimates the difference between groups while quantifying the statistical uncertainty due to individual-to-individual variation and other sources of variability. We validated *scDist* through the unbiased recapitulation of known relationships between immune cells and demonstrated its effectiveness in mitigating false positives from patient-level and technical variations in both simulated and real datasets. Notably, *scDist* facilitates biological discoveries from scRNA cohorts, even when the number of individuals is limited, a common occurrence in human scRNA-seq datasets. We also pointed out how the detection of cell-type specific differences can be obscured by batch effects or other confounders and how the linear model used by our approach permits accounting for these.

Since the same expression data is used for annotation and by *scDist*, there are potential issues associated with "double dipping." Our simulation highlighted this issue by showing that condition-specific effects can result in over-clustering and downward bias in the estimated distances (Methods, Fig. S23). Users can avoid these false negatives by using annotation approaches that can control for

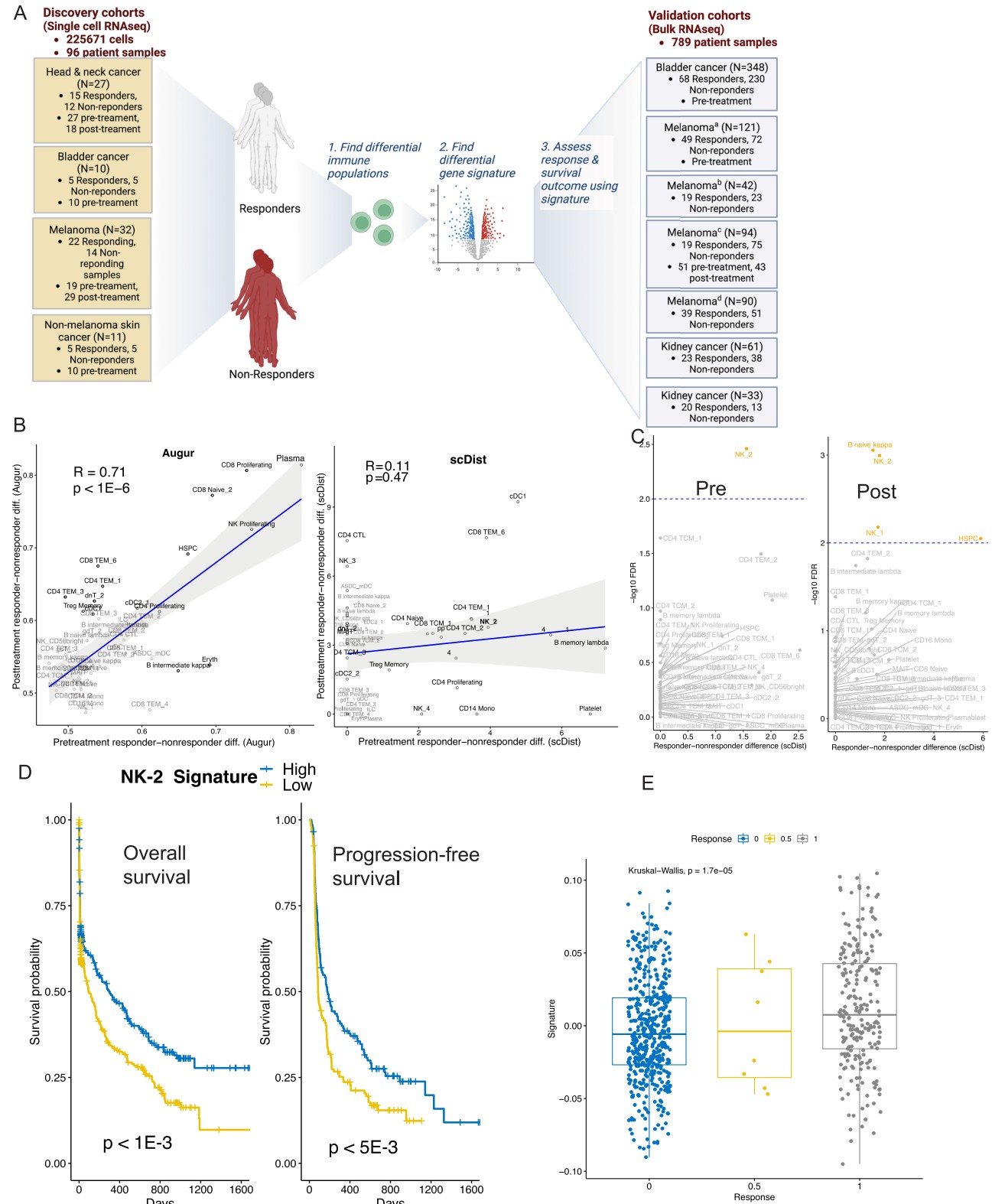

**Fig. 6 | Immunotherapy cohorts analysis using *scDist*. A** Study design: discovery cohorts of four scRNA cohorts (cited in order as shown[2,23–25]) identify cell-type-specific differences and a differential gene signature between responders and non-responders. This signature was evaluated in validation cohorts of six bulk RNA-seq cohorts (cited in order as shown[32–38]). **B** Pre-treatment and post-treatment sample differences were estimated using *Augur* and *scDist* (Spearman correlation is reported on the plot). The error bars represent 95% confidence interval for the fitted linear regression line. **C** Significance of the estimated differences (*scDist*). **D** Kaplan–Meier plots display the survival differences in anti-PD-1 therapy patients, categorized by low-risk and high-risk groups using the median value of the NK-2 signature value; overall and progression-free survival is shown. **E** NK-2 signature levels in non-responders, partial-responders, and responders (Bulk RNA-seq cohorts). The boxplots report the median and first/third quartiles. **A** created with BioRender.com, released under a Creative Commons Attribution-NonCommercial-NoDerivs 4.0 International license. Source data are provided as a Source Data file.

patient and condition-specific effects. *scDist* provides two diagnostic tools to help users identify potential issues in their annotation (Figs. S24 and S6. Despite this, significant errors in clustering and annotation could cause unavoidable bias in *scDist*, and thus designing a cluster-free extension of *scDist* is an area for future work. *scDist* also provides a diagnostic tool that estimates distances at multiple resolutions to help users identify potential issues in their annotation (Fig. S24). Another point of sensitivity for *scDist* is the choice of the number of principal components used to estimate the distance. Although in practice we observed that the distance estimate is stable as the number of PCs varies between 20 and 50 (Fig. S22), an adaptive approach for selecting $K$ could improve performance and maximize power. Finally, although Pearson residual-based normalized counts[12,13] is recommended input for *scDist*, if the data available was normalized by another, sub-optimal, approach, *scDist*'s performances could be affected. A future version could adapt the model and estimation procedure so that *scDist* can be directly applied to the counts, and avoid potential problems introduced by normalization.

We believe that *scDist* will have extensive utility, as the comparison of single-cell experiments between groups is a common task across a range of research and clinical applications. In this study, we have focused on examining discrete phenotypes, such as infected versus non-infected (in COVID-19 studies) and responders vs. non-responders to checkpoint inhibitors. However, the versatility of our framework allows for extension to experiments involving continuous phenotypes or conditions, such as height, survival, and exposure levels, to name a few. As single-cell datasets continue to grow in size and complexity, *scDist* will enable rigorous and reliable insights into cellular perturbations with implications for human health and disease.

## Methods

### Normalization

Our method takes as input a normalized count matrix (with corresponding cell type annotations). We recommend using *scTransform*[13] to normalize, although the method is compatible with any normalization approach. Let $y_{ijg}$ be the UMI counts for gene $1 \le g \le G$ in cell $i$ from sample $j$. *scTransform* fits the following model:

$$y_{ijg} \sim \text{NB}(\mu_g, \alpha_g) \tag{3}$$

$$\log \mu_g = \beta_{0g} + \beta_{1g} \log r_{ij} \tag{4}$$

where $r_{ij}$ is the total number of UMI counts for the particular cell. The normalized counts are given by the Pearson residuals of the above model:

$$z_{ijg} = \frac{y_{ijg} - \hat{\mu}_g}{\sqrt{\hat{\mu}_g + \hat{\mu}_g^2 / \hat{\alpha}_g}} \tag{5}$$

### Distance in normalized expression space

In this section, we describe the inferential procedure of *scDist* for cases without additional covariates. However, the procedure can be generalized to the full model (18) with arbitrary covariates (design matrix) incorporating random and fixed effects, as well as nested-effect mixed models. For a given cell type, we model the $G$-dimensional vector of normalized counts as

$$z_{ij} = \alpha + x_{ij}\beta + \omega_j + \varepsilon_{ij} \tag{6}$$

where $\alpha, \beta \in \mathbb{R}^G$, $x_{ij}$ is a binary indicator of condition, $\omega_j \sim \mathcal{N}(0, \tau^2 I_G)$, and $\varepsilon_{ij} \sim \mathcal{N}(0, \sigma^2 I_G)$. The quantity of interest is the Euclidean distance

between condition means $\alpha$ and $\alpha + \beta$:

$$D := \sqrt{\beta^T \beta} = \|\beta\|_2 \tag{7}$$

If $U \in \mathbb{R}^{G \times G}$ is an orthonormal matrix, we can apply $U$ to equation (6) to obtain the *transformed model*:

$$U z_{ij} = U\alpha + x_{ij}U\beta + U\omega_j + U\varepsilon_{ij} \tag{8}$$

Since $U$ is orthogonal, $U\omega_j$ and $U\varepsilon_{ij}$ still have spherical normal distributions. We also have that

$$(U\beta)^T(U\beta) = \beta^T \beta = D^2 \tag{9}$$

This means that the distance in the transformed model is the same as in the original model. As mentioned earlier, our goal is to find $U$ such that

$$D_K := \sqrt{\sum_{k=1}^{K} (U\beta)_k^2} \approx D \tag{10}$$

with $K \ll G$.

Let $Z \in \mathbb{R}^{n \times G}$ be the matrix with rows $z_{ij}$ (where $n$ is the total number of cells). Intuitively, we want to choose a $U$ such that the projection of $z_{ij}$ onto the first $K$ rows of $U$ ($u_1, \ldots, u_K \in \mathbb{R}^G$) minimizes the reconstruction error

$$\sum_{i=1}^{n} \|z_i - (\mu + v_{i1}u_1 + \cdots + v_{iK}u_K)\|_2^2 \tag{11}$$

where $\mu \in \mathbb{R}^G$ is a shift vector and $(v_{ik}) \in \mathbb{R}^{n \times K}$ is a matrix of coefficients. It can be shown that the PCA of $Z$ yields the (orthornormal) $u_1, \ldots, u_K$ that minimizes this reconstruction error[26].

### Inference

Given an estimator $\widehat{(U\beta)}_k$ of $(U\beta)_k$, a naive estimator of $D_K$ is given by taking the square root of the sum of squared estimates:

$$\sqrt{\sum_{k=1}^{K} \widehat{(U\beta)}_k^2}. \tag{12}$$

However, this estimator can have significant upward bias due to sampling variability. For instance, even if the true distance is 0, $\widehat{(U\beta)}_k$ is unlikely to be exactly zero, and that noise becomes strictly positive when squaring.

To account for this, we apply a post-hoc Bayesian procedure to the $\widehat{U\beta}_k$ to shrink them towards zero before computing the sum of squares. In particular, we adopt the spike slab model of[14]

$$\widehat{(U\beta)}_k \sim \mathcal{N}\left((U\beta)_k, \text{Var}\left[\widehat{(U\beta)}_k\right]\right) \tag{13}$$

$$(U\beta)_k \sim \pi_0 \delta_0 + \sum_{t=1}^{T} \pi_t \mathcal{N}(0, \tau_t) \tag{14}$$

where $\text{Var}[\widehat{(U\beta)}_k]$ is the variance of the estimator $\widehat{(U\beta)}_k$, $\delta_0$ is a point mass at 0, and $\pi_0, \pi_1, \ldots \pi_T$ are mixing weights (that is, they are non-negative and sum to 1).[14] provides a fast empirical Bayes approach to estimate the mixing weights and obtain posterior samples of $(U\beta)_k$. Then samples from the posterior of $D_K$ are obtained by applying the formula (12) to the posterior samples of $(U\beta)_k$. We then summarize the posterior distribution by reporting the median and other quantiles. Advantage of this particular specification is that the amount of shrinkage depends on the uncertainty in the initial estimate of $(U\beta)_k$.

We use the following procedure to obtain $\widehat{U\beta}_k$:

1. Use the matrix of PCA loadings as a plug in estimator for $U$. Then $Uz_{ij}$ is the vector of PC scores for cell $i$ in sample $j$.
2. Estimate $(U\beta)_k$ by using lme4[27] to fit the model (6) using the PC scores corresponding to the $k$-th loading (i.e., each dimension is fit independently).

Note that only the first $K$ rows of $U$ need to be stored.

We are particularly interested in testing the null hypothesis of $D_K = 0$ against the alternative $D_d > 0$. Because the null hypothesis corresponds to $(U\beta)_k = 0$ for all $1 \le k \le d$, we can use the sum of individual Wald statistics as our test statistic:

$$W = \sum_{k=1}^{K} W_k = \sum_{k=1}^{K} \left( \frac{\widehat{(U\beta)}_k}{\widehat{se}\left[\widehat{(U\beta)}_k\right]} \right)^2 \qquad (15)$$

Under the null hypothesis that $(U\beta)_k = 0$, $W_k$ can be approximated by a $F_{\nu_k,1}$ distribution. $\nu_k$ is estimated using Satterthwaite's approximation in lmerTest. This implies that

$$W \sim \sum_{k=1}^{K} F_{\nu_k,1} \qquad (16)$$

under the null. Moreover, the $W_k$ are independent because we have assumed that covariance matrices for the sample and cell-level noise are multiples of the identity. Equation (16) is not a known distribution but quantiles can be approximated using Monte Carlo samples. To make this precise, let $W_1, \ldots, W_M$ be draws from equation (16), where $M = 10^5$ and let $W^*$ be the value of equation (15) (i.e., the actual test statistic). Then the empirical $p$-value[28] is computed as

$$\frac{\sum_{i=1}^{M} I(W_i > W^*) + 1}{M + 1} \qquad (17)$$

### Controlling for additional covariates

Because scDist is based on a linear model, it is straightforward to control for additional covariates such as age or sex of a patient in the analysis. In particular, model (18) can be replaced with

$$z_{ij} = \alpha + x_j\beta + \sum_{k=1}^{p} w_{ijk}\gamma_k + \omega_j + \varepsilon_{ij} \qquad (18)$$

where $w_{ijk} \in \mathbb{R}$ is the value of the $k$th covariate for cell $i$ in sample $j$ and $\gamma_k \in \mathbb{R}^G$ is the corresponding gene-specific effect corresponding to the $k$th covariate.

### Choosing the number of principal components

An important choice in scDist is the number of principal components $d$. If $d$ is chosen too small, then estimation accuracy may suffer as the first few PCs may not capture enough of the distance. On the other hand, if $d$ is chosen too large then the power may suffer as a majority of the PCs will simply be capturing random noise (and adding to degrees of freedom to the Wald statistic). Moreover, it is important that $d$ is chosen a priori, as choosing the $d$ that produces the lowest $p$ values is akin to $p$-hacking.

If the model is correctly specified then it is reasonable to choose $d = J - 1$, where $J$ is the number of samples (or patients). To see why, notice that the mean expression in sample $1 \le j \le J$ is

$$x_j\beta + \omega_j \in \mathbb{R}^G \qquad (19)$$

In particular, the $J$ sample means lie on a $(J-1)$-dimensional subspace in $\mathbb{R}^G$. Under the assumption that the condition difference and sample-level variability is larger than the error variance $\sigma^2$, we should expect that the first $J - 1$ PC vectors capture all of the variance due to differences in sample means.

In practice, however, the model can not be expected to be correctly specified. For this reason, we find that $d = 20$ is a reasonable choice when the number of samples is small (as is usually the case in scRNA-seq) and $d = 50$ for datasets with a large number of samples. This is line with other single-cell methods, where the number of PCs retained is usually between 20 and 50.

### Cell type annotation and "double dipping"

scDist takes as input an annotated list of cells. A common approach to annotate cells is to cluster based on gene expression. Since scDist also uses the gene expression data to measure the condition difference there are concerns associated with "double-dipping" or using the data twice. In particular, if the condition difference is very large and all of the data is used to cluster it is possible that the cells in the two conditions would be assigned to different clusters. In this case scDist would be unable to estimate the inter-condition distance, leading to a false negative. In other words, the issue of double dipping could cause scDist to be more conservative. Note that the opposite problem occurs when performing differential expression between two estimated clusters; in this case, the $p$-values corresponding to genes will be anti-conservative[29].

To illustrate, we simulated a normalized count matrix with 4000 cells and 1000 genes in such a way that there are two "true" cell types and a true condition distance of 4 for both cell types (Fig. S23a). To cluster (annotate) the cells, we applied $k$-means with various choices of $k$ and compared results by taking the median inter-condition distance across all clusters. As the number of clusters increases, the median distance decays towards 0, which demonstrates that scDist can produce false negatives when the data is over-clustered (Fig. S23b). To avoid this issue, one possible approach is to begin by clustering the data for only one condition and then to assign cells in the other condition by finding the nearest centroid in the existing clusters. When applied to the simulated data this approach is able to correctly estimate the condition distance even when the number of clusters $k$ is larger than the true value.

On real data, one approach to identify possible over-clustering is to apply scDist at various cluster resolutions. We used the expression data from the small COVID-19 data[1] to construct a tree $\mathcal{T}$ with leaf nodes corresponding to the cell types in the original annotation provided by the authors (Fig. S24, see Appendix A for a description of how the tree is estimated). At each internal node $v \in \mathcal{T}$, we applied scDist to the cluster containing all children of $v$. We can then visualize the estimated distances by plotting the tree (Fig. S24). Situations where the child nodes have a small distance but the parent node has a large distance could be indicative of over-clustering. For example, PB cells are almost exclusiviely found in cases (1977 cells in cases and 86 cells in controls), suggesting that it is reasonable to consider PB and B cells as a single-cell type when applying scDist.

### Feature importance

To better understand the genes that drive the observed difference in the CD14+ monocytes, we define a gene importance score. For $1 \le k \le d$ and $1 \le g \le G$, the $k$-th importance score for gene $g$ is $|U_{kg}|\beta_g$. In other words, the importance score is the absolute value of the gene's $k$-th PC loading times its expression difference between the two conditions. Note that the gene importance score is 0 if and only if $\beta_g = 0$ or $U_{kg} = 0$. Since the $U_{kg}$ are fixed and known, significance can be assigned to the gene importance score using the differential expression method used to estimate $\beta_g$.

### Simulated single-cell data

We test the method on data generated from model equation (6). To ensure that the "true" distance is $D$, we use the R package uniformly[30] to

draw $\beta$ from the surface of the sphere of radius $D$ in $\mathbb{R}^G$. The data in Figs. 1C and 3C are obtained by setting $\beta = 0$ and $\sigma^2 = 1$ and varying $\tau^2$ between 0 and 1.

## Weighted distance

By default, *scDist* uses the Euclidean distance $D$ which treats each gene equally. In cases where a priori information is available about the relevance of each gene, *scDist* provides the option to estimate a weighted distance $D_w$, where $w \in \mathbb{R}^G$ has non-negative components and

$$D_w = \sum_{g=1}^{G} w_g \beta_g^2 \qquad (20)$$

The weighted distance can be written in matrix form by letting $W \in \mathbb{R}^{G \times G}$ be a diagonal matrix with $W_{gg} = w_g$, so that

$$D_w = \boldsymbol{\beta}^\top W \boldsymbol{\beta} \qquad (21)$$

Thus, the weighted distance can be estimated by instead considered the transformed model where $U\sqrt{W}$ is applied to each $z_{ij}$. After this different transformed model is obtained, estimation and inference of $D_w$ proceeds in exactly the same way as the unweighted case.

To test the accuracy of the weighted distance estimate, we considered a simulation where each gene had only a 10% chance of having $\beta_g \neq 0$ (otherwise $\beta_g \sim \mathcal{N}(0,1)$). We then considered three scenarios: $w_g = 1$ if $\beta_g \neq 0$ and $w_g = 0$ otherwise (correct weighting), $w_g = 1$ for all $g$ (unweighted), and $w_g = 1$ randomly with probability 0.1 (incorrect weights). We then quantified the performance by taking the absolute value of the error between $\sum_g \beta_g^2$ and the estimated distance. Figure S3 shows that correct weighting slightly outperforms unweighted *scDist* but random weights are significantly worse. Thus, the unweighted version of *scDist* should be preferred unless strong a priori information is available.

## Robustness to model misspecification

The *scDist* model assumes that the cell-specific variance $\sigma^2$ and sample-specific variance $\tau^2$ are shared across genes. The purpose of this assumption is to ensure that the noise in the transformed model follows a spherical normal distribution. Violations of this assumption could lead to miscalibrated standard errors and hypothesis tests but should not effect estimation. To demonstrate this, we considered simulated data where each gene has $\sigma_g \sim \text{Gamma}(r, r)$ and $\tau_g \sim \text{Gamma}(r/2, r)$. As $r$ varies, the quality of the distance estimates does not change significantly (Fig. S26).

## Semi-simulated COVID-19 data

COVID-19 patient data for the analysis was obtained from ref. 17, containing 1.4 million cells of 64 types from 284 PBMC samples collected from 196 individuals, including 171 COVID-19 patients and 25 healthy donors.

**Ground truth.** We define the ground truth as the cell-type specific transcriptomic differences between the 171 COVID-19 patients and the 25 healthy controls. Specifically, we used the following approach to define a ground truth distance:

1. For each gene $g$, we computed the log fold changes $L_g$ between COVID-19 cases and controls, with $L_g = E_g(Covid) - E_g(Control)$, where $E_g$ denotes the log-transformed expression data $\log(1 + x)$.
2. The ground truth distance is then defined as $D = \sum_g L_g^2$.

Subsequently, we excluded any cell types not present in more than 10% of the samples from further analysis. For true negative cell types, we identified the top 5 with the smallest fold change and a representation of over 20,000 cells within the entire dataset. When

attempting similar filtering based on cell count alone, no cell types demonstrated a sufficiently large true distance. Consequently, we chose the top four cell types with over 5000 cells as our true positives Fig. S11.

Using the ground truth, we performed two separate simulation analyses:

1: *Simulation analyses I* (Fig. 5A, B): Using one half of the dataset (712621 cells, 132 case samples, 20 control samples), we created 100 subsamples consisting of 5 cases and 5 controls. For each subsample, we applied both *scDist* and *Augur* to estimate perturbation/distance between cases and controls for each cell type. Then we computed the correlation between the ground truth ranking (ordering cells by sum of log fold changes on the whole dataset) and the ranking obtained by both methods. For *scDist*, we restricted to cell types that had a non-zero distance estimate in each subsample, and for *Augur* we restricted to cell types that had an AUC greater than 0.5 (Fig. 5A). For Fig. 5B, we took the mean estimated distance across subsamples for which the given cell type had a non-zero distance estimate. This is because in some subsamples a given cell type could be completely absent.

2: *Simulation analyses II* (Fig. 5C–F): We subsampled the COVID-19 cohort with 284 samples (284 PBMC samples from 196 individuals: 171 with COVID-19 infection and 25 healthy controls) to create 1,000 downsampled cohorts, each containing samples from 10 individuals (5 with COVID-19 and 5 healthy controls). We randomly selected each sample from the downsampled cohort, further downsampled the number of cells for each cell type, and selected them from the original COVID-19 cohort. This downsampling procedure increases both cohort variability and cell-number variations.

*Performance Evaluation in Subsampled Cohorts* : We applied *scDist* and *Augur* to each subsampled cohort, comparing the results for true positive and false positive cell types. We partitioned the sampled cohorts into 10 groups based on cell-number variation, defined as the number of cells in a sample with the highest number of cells for false-negative cell types divided by the average number of cells in cell types. This procedure highlights the vulnerability of computational methods to cell number variation, particularly in negative cell types.

## Analysis of immunotherapy cohorts

**Data collection.** We obtained single-cell data from four cohorts[2,23–25], including expression counts and patient response information.

**Pre-processing.** To ensure uniform processing and annotation across the four scRNA cohorts, we analyzed CD45+ cells (removing CD45− cells) in each cohort and annotated cells using Azimuth[31] with reference provided for CD45+ cells.

**Model to account for cohort and sample variance.** To account for cohort-specific and sample-specific batch effects, *scDist* modeled the normalized gene expression as:

$$Z \sim X + (1|\gamma : \omega) \qquad (22)$$

Here, Z represents the normalized count matrix, X denotes the binary indicator of condition (responder = 1, non-responder = 0); $\gamma$ and $\omega$ are cohort and sample-level random effects, and $(1|\gamma : \omega)$ models nested effects of samples within cohorts. The inference procedure for distance, its variance, and significance for the model with multiple cohorts is analogous to the single-cohort model.

**Signature.** We estimated the signature in the *NK-2* cell type using differential expression between responders and non-responders. To account for cohort-specific and patient-specific effects in differential expression estimation, we employed a linear mixed model described

above for estimating distances, performing inference for each gene separately. The coefficient of *X* inferred from the linear mixed models was used as the estimate of differential expression:

$$Z \sim X + (1|\gamma : \omega) \tag{23}$$

Here, Z represents the normalized count matrix, X denotes the binary indicator of condition (responder = 1, non-responder = 0); $\gamma$ and $\omega$ are cohort and sample-level random effects, and $(1|\gamma: \omega)$ models nested effects of samples within cohorts.

**Bulk RNA-seq cohorts.** We obtained bulk RNA-seq data from seven cancer cohorts[32–38], comprising a total of 789 patients. Within each cohort, we converted counts of each gene to TPM and normalized them to zero mean and unit standard deviation. We collected survival outcomes (both progression-free and overall) and radiologic-based responses (partial/complete responders and non-responders with stable/progressive disease) for each patient.

**Evaluation of signature in bulk RNA-seq cohorts.** We scored each bulk transcriptome (sample) for the signature using the strategy described in ref. 39. Specifically, the score was defined as the Spearman correlation between the normalized expression and differential expression in the signature. We stratified patients into two groups using the median score for patient stratification. Kaplan–Meier plots were generated using these stratifications, and the significance of survival differences was assessed using the log-rank test. To demonstrate the association of signature levels with radiological response, we plotted signature levels separately for non-responders, partial-responders, and responders.

**Evaluating *Augur* Signature in Bulk RNA-Seq Cohorts.** A differential signature was derived for *Augur*'s top prediction, plasma cells, using a procedure analogous to the one described above for *scDist*. This plasma signature was then assessed in bulk RNA-seq cohorts following the same evaluation strategy as applied to the *scDist* signature.

### Statistics and reproducibility
No statistical method was used to predetermine sample size. No data were excluded from the analyses. The experiments were not randomized. The Investigators were not blinded to allocation during experiments and outcome assessment.

### Reporting summary
Further information on research design is available in the Nature Portfolio Reporting Summary linked to this article.

## Data availability
Table 1 gives a list of the datasets used in each figure, as well as details about how the datasets can be obtained. Source data are provided with this paper.

## Code availability
*scDist* is available as an R package and can be downloaded from GitHub[40]: github.com/phillipnicol/scDist. The repository also includes scripts to replicate some of the figures and a demo of *scDist* using simulated data.

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

## Acknowledgements

P.B.N. is supported by NIH T32CA009337. A.D.S. received support from R00CA248953, the Michelson Foundation, and was partially supported by the UNM Comprehensive Cancer Center Support Grant NCI P30CA118100. We express our gratitude to Adrienne M. Luoma, Shengbao Suo, and Kai W. Wucherpfennig for providing the scRNA data[23]. We also thank Zexian Zeng for assistance with downloading and accessing the bulk RNA-seq dataset.

## Author contributions

P.B.N., D.P., G.Q., X.S.L., R.I., and A.D.S. conceived the study. P.B.N. and A.D.S. implemented the method and performed the experiments. P.B.N., R.I., and A.D.S. wrote the manuscript.

## Competing interests

X.S.L. conducted the work while being on the faculty at DFCI, and is currently a board member and CEO of GV20 Therapeutics. P.B.N., D.P., G.Q., R.I., and A.D.S. declare no competing interests.
