## [Peer Review File · Nature Communications]

Robust identification of perturbed cell types in single-cell RNA-seq dataEditorial Note: This manuscript has been previously reviewed at another journal that is not operating a transparent peer review scheme. This document only contains reviewer comments and rebuttal letters for versions considered at *Nature Communications*.

Reviewer #1 (Remarks to the Author):

This is a revised manuscript for a previous submission. I have evaluated the authors' response to the original reviewer 1's questions. Overall, the authors have done a good job and have addressed the original questions. I have several additional comments and hope that they can help improve the manuscript further.

1) In formula 1, all genes are assumed to have common variance τ^2 and σ^2 . In formula 2, expression differences of different genes are treated equally (i.e. if one writes $D^2 = \sum(w_g \cdot \beta_g^2)$, then the weight of each gene $w_g=1$ and it is the same for all genes. In reality, genes are highly heterogeneous. Their variances are most likely unequal. The beta of one gene may not be directly comparable to the beta of another gene, and the optimal distance D may require gene-specific weights to calibrate beta across genes. It will be useful to discuss implications of these assumptions and how they could affect one's analysis and conclusions.

2) Figure S20 legend: did you use 10 samples or 15 samples? The description is inconsistent.

3) In Figure S8a, pDC is in gray, indicating that it does not pass the significance cutoff. Why do you claim that scDist identified pDC as a perturbed cell type not identified in the original study?

Reviewer #2 (Remarks to the Author):

In this revised version of their manuscript the authors have made substantial inclusions of discussions regarding the cell type annotation strategies necessary for the use of their method. This was the main concern in the previous version — in the version a number of simulation studies are supporting the recommendations made by the authors regarding cell type annotation.

Remaining, less concerning, points, have also been addressed.

Reviewer #3 (Remarks to the Author):

The authors addressed many comments raised by the reviewers. Their additional analyses demonstrated the usefulness of the method in some settings when compared to DE based analysis. They have also added discussion about some underlying assumptions, most notably cell type assignment as part of the input. Finally, they greatly clarified the main statistical / computational contribution and provided a much better description for where the method should be considered with respect to prior work for sc and bulk data.

However, the reviewer is not convinced that some of the major concerns (including those raised by multiple reviewers) were satisfactorily addressed in this new version.

1. Several relevant publications / methods were mentioned in the original review though apart from stating that mixed effects are not the focus the authors did not seem to address the request for comparison. Even the comparison they did add, to DE analysis, is inconclusive. It indeed shows that if very few cells from a specific type are present (200 in the current analysis) scDist improves over DE. But this is somewhat unusual. Most studies have many more cells from each type (with the total number of cells in the tens of thousands). So how significant is the improvement for practical analysis?

2. Fig S20 is somewhat confusing since both the response and the caption indicate two different sizes that are considered (10 and 15). Which one is it? And what about much larger sets? The COVID dataset contained close to 200 individuals. What happens when considering bigger groups (100 for example)? How do the comparisons for both the DE and Augur change?

3. The new paper is clearer in terms of the novelty and focus. However, this also raises new questions about the biological relevance of the metric being used for determining cell type

divergence (sum of squares). Why is this a relevant / reasonable way to compare cell type drift? Would be good to justify the use of the method not just in terms of results but also in terms of motivation.

4. The issue of cell type assignment is addressed in the revised manuscript. The idea to use the method to infer if cell types are mistakenly separated in a joint analysis is interesting and actually seem like an interesting direction in its own right. However, it is still not clear to me how this will address the main concern. If the drift is big enough the assignment will likely be the step in which it would have an impact (either assigning to the wrong cell type for one of the conditions or regarding it as a new cell type). The current solution can help though this obviously requires more parameters from the users (definition of distance).

Response to reviews for “Robust identification of perturbed cell types in single-cell RNA-seq data”

Introduction and Summary of major changes

We were pleased to hear that all three reviewers agreed that our previous revisions improved the clarity and quality of the manuscript. We also thank the reviewers for providing additional feedback and insights that further strengthen the scDist method. We begin by briefly summarizing the major changes made to the manuscript:

- As a reviewer pointed out, it may be beneficial to weight genes differently in the definition of scDist. This inspired us to add the possibility of inferring a weighted distance to the scDist R package. Our revised manuscript contains examples on how the weighted distance can improve accuracy when the correct a priori weighting is known.
- Our revised manuscript greatly expands the comparison to counting the number of differentially expressed genes. As we mention in the introduction, this measure of perturbation is extremely sensitive to quantities that increase the power of the corresponding hypothesis test, such as the number of cells or samples. Our updated analyses make this fundamental limitation clear using realistic data.
- Our previous revision showed that bias in scDist can occur when the condition-specific perturbation is too large. Our updated manuscript adds an additional diagnostic tool to help users check for this in their data. Specifically, we compare the inter-condition distance to the inter-cell-type distance, as potential bias can occur when the former is close to or larger than the latter.

Below we have included a response to all of the reviewer comments. The reviewer comments are shown in blue. The changes made in the manuscript (main text or supplementary notes) are shown in red. Our responses are in black. References cited in any responses are collected in a single list at the end of this document.

Response to comments from Reviewer # 1

This is a revised manuscript for a previous submission. I have evaluated the authors’ response to the original reviewer 1’s questions. Overall, the authors have done a good job and have addressed the original questions. I have several additional comments and hope that they can help improve the manuscript further.

Response: We thank the reviewer for encouraging comments.

1. In formula 1, all genes are assumed to have common variance τ^2 and σ^2 . In formula 2, expression differences of different genes are treated equally (i.e. if one writes $D^2 = \sum(w_g * \beta_g^2)$, then the weight of each gene $w_g = 1$ and it is the same for all genes. In reality, genes are highly heterogeneous. Their variances are most likely unequal. The beta of one gene may not be directly comparable to the beta of another gene, and the optimal distance D may require gene-specific weights to calibrate beta across genes. It will be useful to discuss implications of these assumptions and how they could affect one’s analysis and conclusions.

Response: We thank the reviewer for raising this important issue. First, we have conducted a new simulation analysis to show the implication of the assuming τ^2 and σ^2 are shared across genes. The assumption was

made only used to argue that the models for different PCs should have independent noise. Violations of this assumption could lead to miscalibrated standard errors, but should not affect the actual estimate of distance. To demonstrate this, we simulated data where each gene had an individual (random) σ_g and τ_g . Figure S26 shows that the error of the estimation does not depend strongly on the variance of the distribution from which σ_g and τ_g were sampled.

Figure S26: Robustness of *scDist* distance estimates to the shared τ and σ parameters. We simulated datasets where each gene has separate (random) σ_g and τ_g . Specifically, we drew $\sigma_g \sim \text{Gamma}(r, r)$ and $\tau_g \sim \text{Gamma}(r/2, r)$ so that $\text{Var}(\sigma_g) = 1/r$. As r varies, the estimation accuracy of the distance D does not change significantly.

Note that we also recommend using the *scTransform* Hafemeister and Satija (2019) model for normalizing the counts prior to input to *scDist*:

$$z_{ijg} = \frac{y_{ijg} - \hat{\mu}_g}{\sqrt{\hat{\mu}_g + \hat{\mu}_g^2 / \hat{\alpha}_g}}$$

Since the denominator is the variance under a negative binomial distribution, the normalized counts for each gene should have a variance close to 1. Similarly, when other normalization methods are used, such as $\log(1+x)$, genes should be scaled to have mean 0 and standard deviation 1. When this is the case, β_g can be interpreted as the number of standard deviations (for that particular gene) between the condition means.

Second, following up on the reviewer’s suggestion to consider a weighted distance, we have now added this feature to *scDist*. Specifically, we aim to estimate

$$D_w = \sum_{g=1}^G w_g \beta_g^2$$

This model will be useful in cases where certain genes are more likely to contribute to phenotypic changes, so that the weights corresponding to these genes could be larger than others. We extended *scDist* to account for user-defined weights (see below for technical details). Note that similar to a Bayesian approach where uninformative priors are recommended unless there is a priori information, it is recommended to use uniform weights. This is because distance estimates could be biased when the wrong weights are used. To show this, we conducted a simulation to illustrate this sensitivity as well as test the accuracy of this approach. Specifically, we constructed a simulated cell type where only 100 out of the 1000 ground truth β_g are non-zero. In this case, better estimation is achieved by setting $w_g = 1$ for ground truth genes and $w_g = 0$ for others (Fig S3). However, when the incorrect weights are used, the performance is significantly worse than the unweighted version of *scDist* (Fig S3). Thus, the unweighted (default) version of *scDist* should be preferred unless strong a priori information is available.

The relevant figures have been added to the supplementary material and the following paragraph has been added to Results section:

“Though the *scDist* distance D assigns each gene equal weight (unweighted), *scDist* includes an option to assign different weights w_g to each gene (Methods). Weighting could be useful in situations where certain genes are known to contribute more to specific phenotypes. We conducted a simulation to study impact of using the weighted distance. These simulations show that when a priori information is available, using the correct weighting leads to slightly better estimation of the distance. However, incorrect weighting leads to significantly worse estimation compared to the unweighted distance (Fig S3). Therefore, the unweighted distance is recommended unless strong a priori information is available.”

The following section detailing the weighted distance and robustness to model assumptions were added to the Methods section:

“**Weighted distance.** By default, *scDist* uses the Euclidean distance D which treats each gene equally. In cases where a priori information is available about the relevance of each gene, *scDist* provides the option to estimate a weighted distance D_w , where $w \in \mathbb{R}^G$ has non-negative components and

$$D_w = \sum_{g=1}^G w_g \beta_g^2 \tag{1}$$

The weighted distance can be written in matrix form by letting $W \in \mathbb{R}^{G \times G}$ be a diagonal matrix with $W_{gg} = w_g$, so that

$$D_w = \beta^T W \beta \tag{2}$$

Thus, the weighted distance can be estimated by instead considered the transformed model where $U\sqrt{W}$ is applied to each z_{ij} . After this different transformed model is obtained, estimation and inference of D_w proceeds in exactly the same way as the unweighted case.

To test the accuracy of the weighted distance estimate, we considered a simulation where each gene had only a 10% chance of having $\beta_g \neq 0$ (otherwise $\beta_g \sim \mathcal{N}(0, 1)$). We then considered three scenarios: $w_g = 1$ if $\beta_g \neq 0$ and $w_g = 0$ otherwise (correct weighting), $w_g = 1$ for all g (unweighted), and $w_g = 1$ randomly with probability 0.1 (incorrect weights). We then quantified the performance by taking the absolute value of the error between $\sum_g \beta_g^2$ and the estimated distance. Figure S3 shows that correct weighting slightly outperforms unweighted *scDist* but random weights are significantly worse. Thus, the unweighted version of *scDist* should be preferred unless strong a priori information is available.

Robustness to model misspecification. The *scDist* assumes that the cell-specific variance σ^2 and sample-specific variance τ^2 are shared across genes. The purpose of this assumption is to ensure that the noise in the transformed model follows a spherical normal distribution. Violations of this assumption could

Figure S3: Simulation demonstrating the weighted version of *scDist*. 100 simulated datasets are generated with $\beta_g = 0$ with probability 0.9 and $\beta_g \sim \mathcal{N}(0, 1)$ otherwise. Unweighted sets $w_g = 1$ for all g (this is the default mode of *scDist*). Correct weights sets $w_g = 1$ if $\beta_g \neq 0$ and $w_g = 0$ otherwise. Random weights sets $w_g = 1$ with probability 0.1 randomly. Performance is assessed by taking the absolute value of the difference between $\sum_g \beta_g^2$ and the estimated distance.

lead to miscalibrated standard errors and hypothesis tests but should not effect estimation. To demonstrate this, we considered simulated data where each gene has $\sigma_g \sim \text{Gamma}(r, r)$ and $\tau_g \sim \text{Gamma}(r/2, r)$. As r varies, the quality of the distance estimates do not change significantly (**Fig S26**)."

Figure S25. the subsampling analysis with more patients. Figure 5a and Figure 5b used 5 samples per condition. **Top:** Repeating the analysis with 10 subsamples per patient. **Bottom:** With 15 subsamples, the average spearman correlation is higher (0.44 vs 0.54), the mean width of the error bars is lower (6.01 vs 4.68), and the correlation with the ground truth distance is slightly higher (0.30 vs 0.39).

2. Figure S20 legend: did you use 10 samples or 15 samples? The description is inconsistent.

Response: We thank the reviewer for pointing out this omission. The figure has been corrected to indicate to indicate which plots correspond to 10 subsamples and which correspond to 15 subsamples. The correction has been made to Figure S25 in the main text.

3. In Figure S8a, pDC is in gray, indicating that it does not pass the significance cutoff. Why do you claim that scDist identified pDC as a perturbed cell type not identified in the original study?

Response: We thank the reviewer for pointing out this issue. We note that the distance estimate for pDC cells are 5.14 with $p = 0.045$. However, after adjusting for multiple comparisons the p-value for pDC was 0.107, just below the FDR threshold of 0.1 (Figure S13). The combination of the relatively large distance estimate, the uncorrected p -value, and the limited number of patients in this particular dataset (13 total with 6 and 7 in two conditions) suggests that this cell type is likely perturbed. We have clarified the text to note that the pDC result does not persist after false discovery adjustment:

“scDist also identified two cell-types not considered in the original study, dendritic cells (DCs) and plasmacytoid dendritic cells (pDCs) ($p = 0.01$ and $p = 0.04$, Fig S13a), although pDC did not remain significant after adjusting for multiple testing.”

Response to comments from Reviewer # 2

In this revised version of their manuscript the authors have made substantial inclusions of discussions regarding the cell type annotation strategies necessary for the use of their method. This was the main concern in the previous version — in the version a number of simulation studies are supporting the recommendations made by the authors regarding cell type annotation.

Remaining, less concerning, points, have also been addressed.

Response: We thank the reviewer for the kind comments and again for the previous feedback that greatly improved the quality of the manuscript.

Response to comments from Reviewer # 3

The authors addressed many comments raised by the reviewers. Their additional analyses demonstrated the usefulness of the method in some settings when compared to DE based analysis. They have also added discussion about some underlying assumptions, most notably cell type assignment as part of the input. Finally, they greatly clarified the main statistical / computational contribution and provided a much better description for where the method should be considered with respect to prior work for sc and bulk data.

However, the reviewer is not convinced that some of the major concerns (including those raised by multiple reviewers) were satisfactorily addressed in this new version.

Response: We thank the reviewer for taking the time to review our manuscript. Our revised manuscript has added several analyses to address the remaining concerns regarding comparison to differential expression and cell type assignment.

1. Several relevant publications / methods were mentioned in the original review though apart from stating that mixed effects are not the focus the authors did not seem to address the request for comparison. Even the comparison they did add, to DE analysis, is inconclusive. It indeed shows that if very few cells from a specific type are present (200 in the current analysis) scDist improves over DE. But this is somewhat unusual. Most studies have many more cells from each type (with the total number of cells in the tens of thousands). So how significant is the improvement for practical analysis?

Response: We appreciate the opportunity to address the concerns raised by the reviewer. Below, we expand our comparisons of scDist with other methods and perform new analyses to demonstrate the significance of scDist's improvement over DE in practice.

First, we expand our comparisons with the existing published methods outlined in the original review. It is imperative to clarify that **the primary objective of scDist, which is the quantitative measurement of cell perturbations, is fundamentally distinct from the objectives targeted by these cited methods.** Here, we further elucidate the distinctions, and perform comparisons wherever applicable for each publication:

1. Yu et al. (2019) BMC Bioinformatics (PMID: 31861977): This article extends limma (Ritchie et al., 2015) framework by incorporating a variance shrinkage approach (FMT) for linear-mixed effects models. In particular, this article proposed adjusting degrees of freedom for improved statistical power in a t -test for differential expression. However, this method was developed for bulk RNA-seq data, where the limited number of observations necessitates variance shrinkage to increase statistical power. Moreover, while FMT was designed to be applied directly to log-transformed expression data, scDist is applied to the PCA scores. Nonetheless, we assessed FMT's applicability to single cell data within the scDist model. We revisited the analysis of Figure 3b, incorporating p -value calculation using the FMT method (Fig S27). As the data consists no biological differences, this analysis estimated the type I error rate of the hypothesis test that the distance is 0. The type I error was 0.28, substantially exceeding the nominal level of 0.05, suggesting that the FMT may yield positive rates in single cell context due to inflated p -values.

The description of the above analysis was placed in Appendix C.

Figure S27: Repeating the analysis of Figure 3b using the FMT method Yu et al. (2019) instead of lmerTest. The dashed red line indicates the $p = 0.05$ cutoff.

2. Trabzuni et al. Bioinformatics 2014 (PMID: 24519379): This article considered the following mixed model for microarray data:

$$Expression \sim \mu + (1|Array) + (1|Transcript) + (1|Transcript : Condition)$$

We attempted to adapt this model to measure perturbation in single-cell data by estimating the standard deviation of the interaction term $(1|Transcript : Condition)$ (here transcript is replaced with the gene name). We then applied this approach to analyze the dataset where healthy patients were divided arbitrarily into two groups (negative control Figure 1). Similar to *Augur*, the method estimated that all cell types have perturbations above the ground truth difference of 0, falsely indicating group-level differences (Fig S28). More importantly, this approach proved computationally prohibitive for single-cell data, taking nearly 40 minutes compared to 40 seconds for scDist on what would be considered a relatively small single-cell dataset (Fig S28). This complexity stems from the potential size of the expression vector in single-cell datasets, which could exceed 10^9 in large datasets (the number of elements is $(\# \text{ of cells}) \times (\# \text{ of genes})$). In contrast, scDist optimizes computational efficiency by reducing dimensionality through PCA before applying the mixed model.

We have placed this comparison in Appendix D.

3. Zimmerman et al. (2021) Nature Communication (PMID: 33531494): The main contribution of this article is a simulation study showing that not controlling for sample-level variability (pseudoreplication) can lead to false positives when testing for differential expression. In particular, the authors found that mixed models (including individual as a random effect) outperformed the pseudo-bulk approach. However, the authors only address differential expression and do not consider the problem of measuring perturbation between cell types.

We have updated the Results section of our manuscript to explicitly reference this study as additional motivation for using the mixed model within scDist:

Figure S28. Top: Repeating the analysis of Figure 1 using the method of Tratzuni et al. (2014) to measure cell type perturbation (D). The dashed line represents the expected zero perturbation. Bottom: The runtime in minutes of scDist and the Tratzuni method across the 20 repetitions.

“To account for individual-to-individual variability, we modeled the vector of normalized counts with a linear mixed-effects model. Mixed models have previously been shown to be successful at adjusting for this source of variability Zimmerman et al. (2021)”

We also considered counting the number of differentially expressed genes using a test based on a linear mixed-effects model, but this approach had the same limitations as other approaches based on differential expression (Fig S9, the details are provided in response to this reviewer’s second comment).

4. Listgarten et al. (2010) PNAS (PMID: 20810919): This article uses a linear mixed model to adjust for both population structure and expression heterogeneity (batch effects) in eQTL studies, aimed at estimating the change in expression associated with a particular SNP. In particular, the method needs genetic data as input and is designed for bulk RNA-seq. Apart from the utilization of a linear mixed-effects model, there are no substantial similarities between the Listgarten et al. approach and our method, scDist. scDist is specifically designed for single-cell data.

Next, we address the comparison between scDist and counting the number of differentially expressed genes (nDEG). nDEG is overtly sensitive to sample size because this directly increases the power to detect a differentially expressed gene. We demonstrate this by expanding our previous simulation, by sampling cells with replacement from the Wilk et al. (2020) data. As the number of cells increases to 10,000, the number of differentially expressed genes within CD14 monocytes increases (Fig 2a). On the other hand, the scDist distance estimate remains constant (Fig 2b). This is because scDist is attempting to estimate a population quantity (the distance), which does not depend on the sample size. Thus, the number of differentially expressed genes can be a misleading measure of perturbation if there are significant differences in the number of cells per cell type. Heterogeneity in the cell type size distribution is common in real single-cell datasets; for example, in the Wilk et al. (2020) data, the cell-type sizes vary by 2 orders of magnitude (from 234 pDC cells to 10339 CD14 monocytes).

An additional limitation of nDEG is that it requires fitting a model to all G genes, whereas scDist operates on $K \ll G$ principal components. This means that for large datasets nDEG will be much slower. In fact, scDist was over 60 times faster than nDEG when there were 10,000 cells per cell type (Fig S4).

We have updated the manuscript by adding a section and main figure comparing scDist to nDEG. Note that we make several additional comparisons demonstrating the benefit of using scDist over nDEG that will be described in the response to subsequent comments. The following section has been added to the main text:

Comparison to counting the number of differentially expressed genes: We also compared scDist to the approach of counting the number of differentially expressed genes (nDEG) on pseudobulk samples Crowell et al. (2020). Given that the statistical power to detect differentially expressed genes is heavily reliant on sample size, we hypothesized that nDEG could become a misleading measure of perturbation in single-cell data with a large variance in the number of cells per cell type. To demonstrate this, we applied both methods to resampled COVID-19 data Wilk et al. (2020) where the number of cells per cell type was artificially varied between 100 and 10,000. nDEG was highly confounded the number of cells (Fig 2a), whereas the scDist distance remained relatively constant despite the varying number of cells (Fig 2b). When the number of subsampled cells is small, the ranking of cell types (by perturbation) was preserved by scDist but not by nDEG (Fig S5a-c). Additionally, scDist was over 60 times faster than nDEG since the latter requires testing all G genes as opposed to $K \ll G$ PCs (Fig S4).

Figure 2: The number of differentially expressed genes is susceptible to differences in statistical power. **A)** Sampling with replacement from the COVID-19 dataset Wilk et al. (2020) to create datasets with a fixed number of cells per cell type, and then counting the number of differentially expressed genes (nDEG) for the CD14 monocytes. **B)** Repeating the previous analysis with the scDist distance. **C)** Comparing all pairs of cell types on the down-sampled Zheng et al. (2017) dataset and applying hierarchical clustering to the pairwise perturbations. Leaves corresponding to T cells are colored blue while the leaf corresponding to monocytes is colored red. **D)** The same analysis using the scDist distances.

Figure S4: The average runtime (in minutes) of scDist and nDEG (muscat) as the number of cells per cell type varies on the resampled data (see Fig 2a).

2. Fig S20 is somewhat confusing since both the response and the caption indicate two different sizes that are considered (10 and d15). Which one is it? And what about much larger sets? The COVID dataset contained close to 200 individuals. What happens when considering bigger groups (100 for example)? How do the comparisons for both the DE and Augur change?

Response: Thank you for pointing out this omission in Figure S25, and we apologize for any confusion caused by this problem. The figure has been updated to clearly differentiate between the results obtained with 10 and 15 subsamples (see also comment 2 from Reviewer # 1).

Regarding your question on analyzing datasets with more samples, such as those with approximately 100 individuals, we extended our simulations. The updated simulation now includes scenarios with up to 100 patients per group, testing the scalability and robustness of scDist, nDEG, and Augur in conditions closer to those encountered in very large-scale studies. As shown in Figure S9a, while scDist estimates remain stable with number of patients, the nDEG tends to increase due to increasing power. Moreover, scDist estimates had higher correlation with the simulated ground truth whereas nDEG's performance degraded with increasing sample size S9b).

Augur's performance was notably inferior compared to both nDEG and scDist. Unlike nDEG, however, the performance of Augur improved with an increase in sample size. This improvement is likely because the machine learning classifier (random forest) used by Augur is prone to misinterpreting patient-specific differences as condition-specific differences in smaller samples (this was previously shown in Figure 1).

The following section has been added to the Results section to describe these updates:

We also considered varying the number of patients on simulated data with a known ground truth. Again, the nDEG (computed using a mixed model, as recommended by Zimmerman et al. (2021)) increases as the number of patients increases whereas scDist remains relatively stable (Fig S9a). Moreover, the correlation between the ground truth perturbation and scDist increases as the number of patients increases (Fig S9b). Augur was also sensitive to the number of samples and had a lower correlation with the ground truth than both nDEG and scDist.

Figure S9: **a** Comparison of scDist, Augur, and nDEG (this time computed using lme4 Bates et al. (2015)) on simulated data as the number of patients increases. **b** The correlation between the ground truth perturbation (distance) as the number of patients increases.

3. The new paper is clearer in terms of the novelty and focus. However, this also raises new questions about the biological relevance of the metric being used for determining cell type divergence (sum of squares). Why is this a relevant / reasonable way to compare cell type drift? Would be good to justify the use of the method not just in terms of results but also in terms of motivation.

Response: Thank you for raising an important point about the biological relevance of the distance metric used by scDist. Here, we provide a comprehensive justification for the use of this metric as a method to compare cell type drift, addressing both analyses and motivation.

1. Demonstration of Biological Relevance. Our Figure 3d was specifically designed to demonstrate the biological relevance of the scDist distance metric. When the scDist distances computed between pairs of cell types in the Wilk et al. (2020) data were clustered, they recapitulated known relationships between immune cells. We updated the following paragraph in the Results section to clarify that the intent of the analysis was to show the relevance of the distance metric:

“The Euclidean distance D measures perturbation by taking the sum of squared differences across all genes. To show that this measure is biologically meaningful, we applied *scDist* to obtain estimated distances between pairs of known cell types in the above dataset and then applied hierarchical clustering to these distances. The resulting clustering is consistent with known relationships driven by cell lineages (Fig 3d). Specifically, Lymphoid cell types T and NK cells clustered together, while B-cells were further apart, and Myeloid cell types DC, monocytes, and neutrophils were close to each other. ”

To further validate the biological relevance of scDist metrics and discount any issues arising from misannotation in single-cell data, we now have conducted additional analyses. We used a dataset consisting of 8 FACS sorted cell types from Zheng et al. (2017) combined into one dataset by Duò et al. (2018). To create variation in number of cells per cell type, we sub-sampled CD14 Monocytes to have $\approx 10\%$ the number of cells as the other cell types. Similar to the analysis of Figure 3d, we estimated the pairwise distance between all pairs of cell types using scDist and nDEG.

The hierarchical clustering (Fig 2e, included in the response to comment 1) for nDEG showed that memory T cells are more similar to CD14 Monocytes than the other T cell types. This is unexpected because memory T cells should be more similar to other T cell types. In contrast, scDist has the expected result of clustering the T cells together and placing the monocytes as an outlier, confirming that distance metric of scDist captures biologically meaningful differences. The following main text was added to describe this analysis:

An additional limitation of nDEG is that it does not account for the magnitude of the differential expression. We illustrated this with a simple simulation that shows the number of DEGs between can be the same (or less) despite a larger transcriptomic perturbation in gene expression space (Fig S7a,b). To demonstrate this on real data, we considered a dataset consisting of 8 sorted immune cell types (originally from Zheng et al. (2017) and combined by Duò et al. (2018)) where scDist and nDEG were applied to all pairs of cell types and the perturbation estimates were visualized using hierarchical clustering. Although both nDEG and scDist performed well when the sample size was balanced across cell types (Fig S8), nDEG provided inconsistent results when the CD14 Monocytes were downsampled to create a heterogeneous cell type size distribution. Specifically, scDist produced the expected result of clustering the T cells together whereas nDEG places the Monocytes in the same cluster as B and T cells (Fig 2c,d) despite the fact that these belong to different lineages. Thus by taking into account the magnitude of the differential expression, scDist is able to produce results more in line with known biology.

2. Weighted Distance Metric We have also considered a new distance measure in the revised manuscript. Based on Reviewer #1’s suggestion, we investigated *weighted* sum of squares instead of the usual unweighted sum of squares as a scDist distance metric (see our response to the first comment of Reviewer #1) . Indeed, when weights are known, the weighted distance improved accuracy over the unweighted distance (S3). However, using incorrect weights leads to significantly worse performance. Thus, we recommend an unweighted sum of squares unless strong a priori biological knowledge is available. We have added the optional feature of weighted distance to the scDist R package.

Figure S6: Comparing the inter-condition distance (red stars) to the inter-cell-type scDist distance estimates (boxplot) on the Wilk et al. (2020) COVID-19 data.

3. Computational Efficiency A final reason for using the Euclidean distance is that it is the only distance that is invariant to rotation. Because PCA is a rotation of the original coordinate system, the distance in the PC space will be equivalent to the original distance (and approximately equivalent if only the top K PCs are used). This is the reason why scDist is able to compute perturbations significantly faster than the other compared methods.

4. The issue of cell type assignment is addressed in the revised manuscript. The idea to use the method to infer if cell types are mistakenly separated in a joint analysis is interesting and actually seem like an interesting direction in its own right. However, it is still not clear to me how this will address the main concern. If the drift is big enough the assignment will likely be the step in which it would have an impact (either assigning to the wrong cell type for one of the conditions or regarding it as a new cell type). The current solution can help though this obviously requires more parameters from the users (definition of distance).

Response: A figure has been added to demonstrate that within the Wilk et al. (2020) COVID-19 dataset, the inter-condition distances were consistently and significantly smaller than the inter-cell-type distances across all 13 cell types (Fig S6). In general, the comparison of inter-condition distances with inter-cell-type distances can serve as a diagnostic to help determine if scDist findings are free from errors due to issues in the annotation. This diagnostic capability has been added to the scDist R package.

Nevertheless, we agree with the reviewer’s observation that, should the distance between the condition means be exceedingly substantial, it would likely cause an error in the annotation process. In particular, annotation errors could arise when the inter-condition distance is similar (or larger) than the inter-cell-type distance. We have therefore updated the Results section to include the following text:

As potential issues could occur when the inter-condition distance exceeds the inter-cell-type distance, scDist provides a diagnostic plot (Fig S6) to compare these two distances.

When the inter-condition distance is smaller, our previously recommended diagnostic approaches of clustering

using only one condition (Fig S23) or the tree-based analysis (Figure S24) would be likely to flag errors in the annotation.

Note that when there are extreme deviations (inter-condition distance much larger than inter-cell-type distance) due to annotation errors, any method relying on these labels will fail. To make this clear we have also added the following statement to the Discussion:

scDist provides two diagnostic tools to help users identify potential issues in their annotation (Fig S24 and Fig S6). Despite this, significant errors in clustering and annotation could cause unavoidable bias in scDist, and thus designing a cluster-free extension of scDist is an area for future work.

References

- D. Bates, M. Mächler, B. Bolker, and S. Walker. Fitting linear mixed-effects models using lme4. *Journal of Statistical Software*, 67(1):1–48, 2015.
- H. L. Crowell, C. Sonesson, P.-L. Germain, D. Calini, L. Collin, C. Raposo, D. Malhotra, and M. D. Robinson. Muscat detects subpopulation-specific state transitions from multi-sample multi-condition single-cell transcriptomics data. *Nature communications*, 11(1):6077, 2020.
- A. Duò, M. D. Robinson, and C. Sonesson. A systematic performance evaluation of clustering methods for single-cell rna-seq data. *F1000Research*, 7, 2018.
- C. Hafemeister and R. Satija. Normalization and variance stabilization of single-cell rna-seq data using regularized negative binomial regression. *Genome biology*, 20(1):1–15, 2019.
- J. Listgarten, C. Kadie, E. E. Schadt, and D. Heckerman. Correction for hidden confounders in the genetic analysis of gene expression. *Proceedings of the National Academy of Sciences*, 107(38):16465–16470, 2010.
- M. E. Ritchie, B. Phipson, D. Wu, Y. Hu, C. W. Law, W. Shi, and G. K. Smyth. limma powers differential expression analyses for rna-sequencing and microarray studies. *Nucleic acids research*, 43(7):e47–e47, 2015.
- D. Trabzuni, U. K. B. E. C. (UKBEC), and P. C. Thomson. Analysis of gene expression data using a linear mixed model/finite mixture model approach: application to regional differences in the human brain. *Bioinformatics*, 30(11):1555–1561, 2014.
- A. J. Wilk, A. Rustagi, N. Q. Zhao, J. Roque, G. J. Martínez-Colón, J. L. McKechnie, G. T. Ivison, T. Ranganath, R. Vergara, T. Hollis, et al. A single-cell atlas of the peripheral immune response in patients with severe covid-19. *Nature medicine*, 26(7):1070–1076, 2020.
- L. Yu, J. Zhang, G. Brock, and S. Fernandez. Fully moderated t-statistic in linear modeling of mixed effects for differential expression analysis. *BMC bioinformatics*, 20:1–9, 2019.
- G. X. Zheng, J. M. Terry, P. Belgrader, P. Ryvkin, Z. W. Bent, R. Wilson, S. B. Ziraldo, T. D. Wheeler, G. P. McDermott, J. Zhu, et al. Massively parallel digital transcriptional profiling of single cells. *Nature communications*, 8(1):14049, 2017.
- K. D. Zimmerman, M. A. Espeland, and C. D. Langefeld. A practical solution to pseudoreplication bias in single-cell studies. *Nature communications*, 12(1):1–9, 2021.

Reviewer #1 (Remarks to the Author):

The authors have adequately addressed my questions.

A couple of typos:

(1) Line 144: nDEG [was highly confounded the number] of cells.

(2) Line 150: shows the number of nDEGs [between can] be the same

Reviewer #3 (Remarks to the Author):

The authors have successfully addressed my remaining comments

Response to Reviewers

For reference, we have pasted the reviewer comments in blue below:

Reviewer #1 (Remarks to the Author):

The authors have adequately addressed my questions.

A couple of typos:

(1) Line 144: nDEG [was highly confounded the number] of cells.

(2) Line 150: shows the number of nDEGs [between can] be the same

Reviewer #3 (Remarks to the Author):

The authors have successfully addressed my remaining comments

Our response:

We thank the reviewers for taking the time to review our revised manuscript.

We have addressed the two typos that Reviewer #1 identified.

In Line 144, the sentence now correctly reads “nDEG was highly confounded by the number of cells...”

In line 150, the sentence now correctly reads “We illustrated this with a simple simulation that shows the number of DEGs between two cell types can be the same...”